# Caring for People with Rare Diseases: A Systematic Review of the Challenges of, and Strategies for Dealing with, COVID-19

**DOI:** 10.3390/ijerph20196863

**Published:** 2023-09-29

**Authors:** Elena Faccio, Matteo Bottecchia, Michele Rocelli

**Affiliations:** Department of Philosophy, Sociology, Education and Applied Psychology, University of Padua, 35139 Padova, Italy; matteo.bottecchia.2@studenti.unipd.it (M.B.); michele.rocelli@phd.unipd.it (M.R.)

**Keywords:** COVID-19, rare diseases, healthcare, psychological wellness

## Abstract

The COVID-19 pandemic took a toll on everyone’s lives, and patients with rare diseases (RDs) had to pay an even higher price. In this systematic review, we explored the impact of the COVID-19 pandemic on individuals with RDs from a psychological perspective. Using the Preferred Reporting Items for Systematic Reviews and Meta-Analyses (PRISMA) guidelines, we retrieved articles from the Google Scholar, Scopus, and PubMed databases focusing on ‘COVID-19,’ ‘psychology,’ and ‘rare diseases.’ Seventeen primary articles were identified (mainly from continental Europe). The results revealed the psychological effects of the pandemic on rare disease patients, including increased anxiety, stress, and depressive moods. This review also highlighted the increased vulnerability and reduced quality of life of rare disease patients during the pandemic, as well as the importance of telecare and psychological support as critical interventions for improving their well-being. There is an urgent need for multidisciplinary research and stronger healthcare systems to meet the unique challenges of rare disease patients, who represent 3.5–5.9% of the global population.

## 1. Introduction

On 31 December 2019, the Chinese authorities announced a pneumonia outbreak with an unknown cause in Wuhan (Hubei Province, China), and on 11 February 2020, the World Health Organization (WHO) publicised the spread of a respiratory disease named coronavirus disease (COVID-19). On 11 March 2020, after evaluating the global diffusion of the disease, COVID-19 was finally declared a pandemic, which meant that many governments began to implement restrictive policies to limit its spread and protect their populations. These restrictive policies included lockdowns, quarantines, and social distancing, among others, and they were considered necessary to prevent the diffusion of the virus. However, these restrictions also increased people’s loneliness and social isolation, which had repercussions for populations’ physical and mental health [1]. While this was true for almost everyone, it was even truer for people with rare diseases (RDs), their relatives, caregivers, and everyone involved in caring for people with special needs.

There is no uniform definition of what qualifies as a rare disease. Countries around the world have distinct metrics for assessing whether a disease is rare, with ranges of prevalence varying between 5/100,000 cases in Korea and 150/100,000 cases in China. However, a global average of 40 cases per 100,000 people is considered the minimum threshold for a disease to be rare [2]. Many RDs have a prevalence of 1/1,000,000 cases or less, and the Orphanet database counted 6,172 unique RDs in 2019 amongst a conservatively estimated prevalence of 3.5–5.9% of the global population [3], which translates to 280–472 million people affected worldwide. These numbers highlight the importance of scientific research in this field. To put the situation into perspective, Eurostat counted 447.7 million people living in European countries [4], and if one considers that this number intentionally excludes any rare cancers, it is possible to say that people affected by RDs outnumber the population of continental Europe.

We searched the Google Scholar, PubMed, and Scopus databases using ‘COVID-19’, ‘rare diseases’, and ‘psychology’ as keywords, which yielded 217 articles in Google Scholar, 11 in Scopus, and 7 in PubMed. Many of the selected articles examined the impact of COVID-19 on loneliness [5,6,7,8], social isolation [7,9,10], depressive moods [6,7,8,10], distress [7,8,11,12], and anxiety [7,8,9,10,13] in patients with RDs, the people involved with them, and the psychological burden the disease imposed [8,14]. Loneliness and social isolation, although different, are often connected and are already an important social problem, which the pandemic exacerbated [15], especially for the most fragile people in society. Depressive moods, anxiety, and distress surged during the pandemic [16], affecting people’s well-being and placing many at an increased risk of vulnerability. Organisations dealing with RDs faced many challenges during the pandemic due to a lack of personnel, inadequate funding, and concerns about the well-being of patients who could not, or found it increasingly difficult to, access mandatory healthcare programmes or psychological support [17]. To offset some of these challenges, the healthcare community began to implement different kinds of telehealth and telemedicine [10,18,19], which seemed to be viable options that should steadily be introduced into healthcare systems, although they could not address all problems and needed to be broadly implemented to be fully effective [20]. There was also a need for psychological support to help patients with RDs, their caregivers, and their families cope with the changes and difficulties that arise during crises in the clinical and social spheres, and to help them make full use of the resources available [12].

Researchers have made efforts to study and understand the post-COVID-19 period from various perspectives [1,21]. This paper complements the extant research by specifically focusing on SRs and psychology to provide readers with an overview of the challenges, solutions, coping strategies, and peculiarities of the vulnerable but resourceful population with RDs and the people around them.

Therefore, the aim of this review was to raise awareness of RD patients’ psychological well-being during the COVID-19 pandemic, synthesise state-of-the-art research for the scientific community on how health emergencies can be addressed for individuals with RDs, and, hopefully, stimulate more research to underpin the development of effective interventions.

## 2. Materials and Methods

The goal of this review was to study the effects of the COVID-19 pandemic on the lives and psychological well-being of people with rare diseases and all of the people involved with them. Because this review followed the Preferred Reporting Items for Systematic Reviews and Meta-Analyses (PRISMA) guidelines [22], it has not been registered into a database. We searched the Google Scholar, PubMed, and Scopus databases for scientific publications that could give us a broader and deeper view of the challenges that different institutions and people around the world faced during the pandemic and the solutions they developed to offset the emergency and cope with its outcomes. As a first step, a broad search of Google Scholar using the keywords ‘COVID-19,’ ‘psychology,’ and ‘rare diseases’ yielded 2860 reports for 2020–2023, but only 217 of these were scientific papers. The same keywords were then used for a Scopus search within titles, abstracts, and keywords, which yielded 11 articles, most of which overlapped with the research in Google Scholar. In PubMed, the same keywords were used to search within titles, abstracts, and texts, which yielded 7 articles (again largely overlapping with those in Google Scholar). Identical searches were performed on 12 April 2023 and 23 May 2023. The repeated search of Google Scholar yielded 220 articles, but the outputs of the other two databases remained unchanged. We then conducted further selection and filtering using the PRISMA 2020 statement checklist [22].

The screening process involved the first and second authors independently reading the abstracts and texts of the first 102 (47% of the total) articles. After the first screening, the two authors discussed the inclusion criteria, and, considering the focus of this review, they decided to exclude articles that focused only on the medical aspects of the relationship between COVID-19 and RDs. They decided to focus on articles written in English across the world, not limited to a particular RD, and to consider equally salient the points of view of RD patients, their relatives/caregivers, medical professionals, and RD associations. In this phase, 61 articles were excluded due to inconsistencies with the objectives of this paper.

The three authors independently performed a second screening based on the inclusion and exclusion criteria, and another detailed discussion followed this step. The final chosen articles needed to be selected by at least two of the three authors. This screening led to 17 documents that met all of the criteria and another 10 that we considered important for their contribution, topic, or proposed solutions. A flowchart of the literature search is shown in Figure 1. Both research and theoretical documents were deemed important for gaining a broader understanding of the phenomenon and its implications.

The content of, and data from, the chosen articles were analysed using Office 365 Excel (Microsoft Corporation, Redmond, WA, USA) and by focusing on qualitative/quantitative/mixed approaches, data provenance, publication dates, one or multiple RDs, patients/caregivers/families/organisations, and medical or psychological backgrounds. Each of the three authors compiled a collection of highlights from each article to help navigate and synthesise the documents.

For the data extraction and analysis, we decided not to mix statistics regarding different RDs, but to treat them as different entities and present the data according to psychological or social categories.

As this brief explanation of the methodology indicates, one of the major limitations of this systematic review was the lack of scientific publications connecting psychological well-being and RDs with a ‘non-medical-first’ approach. This resulted in a transversal, broad review that was not easy to align with hard statistics, but it nevertheless provided evidence of different approaches, problems, and solutions.

## 3. Results

The studied material consisted mainly of scientific publications (fifteen articles), a thesis dissertation, and one conference paper (for which we could only access the abstract). The geographical distribution of data in the studied material showed a prevalence of research in continental Europe, with 61.1% of the total yield of studies coming from this region. Of these ten documents, five were from England [9,10,14,23,24], three were from Italy [5,7,11], two were from Germany [8,25], and one was from Norway [13]. After Europe, two articles were from North America [12,18], one was from Asia (Hong Kong) [26], one was from the Asia–Pacific region [17] (based on data from Australia, Hong Kong, India, Japan, China, Malaysia, New Zealand, the Philippines, Singapore, and Taiwan), one was from South America (Brazil) [6], and, finally, there was one that took a broad global perspective [19].

Regarding RDs, nine documents approached the theme in a broad, general way [6,8,12,13,14,17,19,25,26], while the rest focused on one or two diseases or on a specific group of diseases per publication. The RDs mentioned in the articles were hereditary haemorrhagic telangiectasia (HTT) [5], lysosomal storage disorder (LSD) [7], Batten disease [18], interstitial cystitis/bladder pain syndrome (IC/BPS) [11], osteogenesis imperfecta [10], congenital surgical diseases [25], oesophageal atresia/tracheo-oesophageal fistula (OA/TOF) [9], rare neurogenetic conditions [24], and macrocytosis [23].

The publications addressed the relationship between COVID-19 and RDs from different perspectives, choosing to focus on patients, families/relatives/caregivers (FRCs), RD organisations, and intervention issues. In particular, seven documents considered the patient’s perspective [6,7,10,12,13,23,26], five focused on FRCs [8,9,14,24,25], four assessed interventions [5,11,18,19], and one examined an RD organisational perspective [17].

As mentioned in the introduction, the psychological studies addressed different issues, such as increases in anxiety, depressive moods, distress, social isolation, loneliness, and the psychological burden for all of the people involved with RD patients. In particular, four articles highlighted stress and distress [7,8,11,12], five anxiety [7,8,9,10,13], four loneliness [5,6,7,8], three social isolation [7,9,10], and four depressive moods [6,7,8,10], but most addressed the psychosocial burden.

It is important to note that because the documents were catalogued according to their content, and many researchers approached the issues from different perspectives, the studies were topic-oriented rather than statistical. Following this brief analysis of the metrics presented in Table 1, this article addresses and explores, section by section, the most interesting topics, starting from the psychological and social points of view of the various documents.

### 3.1. Psychological and Social Points of View

#### 3.1.1. Stress and Distress

The selected papers approached ‘stress’ and ‘distress’ from different perspectives and methodologies across the spectrum of qualitative and quantitative research. To guide the readers, we summarise these conditions as ‘experiencing events with a strong emotional impact’ [27]. The literature highlighted an increase in perceived stress for people with RDs and FRCs alike. In Italy, 66% of participants in Fiumara’s work reported a stressful situation [7], and Rihm found a high stress level amongst almost 90% of caregivers for this group [8]. An important aspect to note is that for many people with RDs, the increase in stress correlated with an increase in health problems, varying from sleep deprivation to worsening disease-related symptoms [8,12].

It is a major concern that stressful situations can worsen an already precarious situation for patients and FRCs, potentially leading to a cascade effect on other aspects of well-being. This is particularly true for already psychologically affected patients, as in the case of interstitial cystitis [11], where distress can cause a decrease in coping capacity. Web-mediated consultations have been recommended to help such patients and their relations. It is interesting to note that some researchers found that during the pandemic, stress levels increased for people with and without RDs in statistically similar ways [7], but the major differences concerned the quality of life baseline of the two groups and the worsening scale of physical consequences.

#### 3.1.2. Anxiety

Many documents reported anxiety as a central aspect or consequence of the pandemic. Interestingly, the Norwegian data showed greater increases in anxiety for people with and without RDs [13], while other research confirmed a substantial equality in anxiety increases between the different studied groups [7]. Analysing the documents, it became clear that the prevalence of anxiety was high [7,10,13] for both patients and caregivers, although sometimes with different meanings. Caregivers of children with RDs often reported anxiety about their future related to the disease, the increased possibility of death, and social isolation [9]. RD patients seemed to experience anxiety in a similar way to people without RDs, but with the aggravation of their specific circumstances. In particular, those with rare neurogenetic conditions associated with anxiety that caused autistic-like behaviour [24], and with papilledema and idiopathic intracranial hypertension, experienced a 64% anxiety increase [28]. All of the documents agreed on the importance of helping people ease their anxiety to enhance their well-being.

#### 3.1.3. Loneliness

Loneliness, also known in the literature as perceived isolation [29], can be defined as a deep feeling of abandonment, emptiness, and alienation, which can become so deep that it threatens people’s well-being [30]. The experience of loneliness is so common in the global population that one study described it as an ‘epidemic in modern society’ [31], so it was not surprising to find it mentioned in many of the studies.

In some cases, people with RDs experience loneliness due to a perceived lack of understanding from other people [11]. According to the articles, during the pandemic, people with RDs and their FRCs experienced some form of loneliness, with prevalence ranging between 49.2% [6] and 70% [7]. Loneliness seemed to be particularly dangerous for people with specific RDs, such as hereditary haemorrhagic telangiectasia, potentially triggering aggression and alcohol and drug abuse. Research has demonstrated the importance of programmes to protect vulnerable people, such as RD patients and the people around them, but the effectiveness of such programmes and their consequences must be evaluated on the medical, ethical, and psychosocial levels [21].

#### 3.1.4. Social Isolation

In contrast to loneliness, social or objective isolation [29] can be described as ‘the inadequate quality and quantity of social relations with other people at the different levels where human interaction takes place’ [32]. Based on this definition, it is possible to consider social isolation as a major by-product of the social distancing mandated by government policies worldwide. These policies basically had two outcomes for people during the pandemic: an increase in familial interactions and a decrease in all others.

Some RD patients began to perceive others negatively and as potentially dangerous to their well-being, although the increase in family interactions and involvement/cooperation was seen as a positive outcome of social distancing [7]. Similarly, social isolation for caregivers, while emotionally taxing, tended to be considered a positive way of protecting their children [9]. Lack of socialisation and peer-to-peer interaction were highlighted in many ways, but most interestingly by the drastic increase in prevalence (from 13% to 83%) in the eight-month span between two iterations of the same survey [10]. This underscores that social isolation increased to an unprecedented magnitude in less than a year. As for loneliness, the lockdowns were an important tool for COVID-19 prevention and, in some cases, for alleviating the burden on FCRs of the preoccupation with the high possibility of infecting loved ones. However, these policies were greatly detrimental to overall well-being, particularly for RD patients [23].

#### 3.1.5. Depressive Moods

A depressive mood is, again, an umbrella term that should not be confused with clinical depression (a major depressive disorder), as defined in DSM-V-TR and ICD-11. The use of this term implies the existence of some symptoms defined in the manuals [33,34], but does not refer to an actual diagnosis.

In the considered articles, it was clear that many depressive moods were experienced, with prevalence ranging from 40.8% in a Brazilian study [6] to 70% in an Italian study [7]. A similar prevalence (69%) was observed in patients with osteogenesis imperfecta in August 2020, which marginally decreased to 61% in the following eight-month period, although it is important to note that the number of participants varied from 110 to 124, translating both percentages to 76 people experiencing depressive moods [10]. Lockdowns negatively influenced people with papilledema and idiopathic intracranial hypertension, who experienced a 51% increase in depressive moods [28]. These figures suggest that the pandemic, as well as the countermeasures, affected the well-being of patients with RDs, shifting their moods negatively and consequently increasing the associated risks. Importantly, RD patients already live challenging lives, and mood changes of this magnitude could easily lead to a greater overall burden.

#### 3.1.6. Psychosocial Burden

A psychosocial burden is a collection of feelings, emotions, and social difficulties related to people’s health-related quality of life, as noted in the literature on cancer [35,36] and other diseases. People with RDs and their FRCs usually live in situations that compromise their physical, social, psychological, and spiritual life balances to some degree. The COVID-19 pandemic added another layer of complexity to these already difficult situations, and while many articles confirmed substantially equal responses to the crisis of people with and without RDs, one should not forget the highly likely difference in their baseline difficulties. The statistics should be read in light of this fact, and consistent countermeasures should be developed to help people with RDs, who constitute 3.5–4.9% of the global population [3].

### 3.2. Points of View of People with RDs, FRCs, and RD Organisations

#### 3.2.1. Experiences of People with RDs

Some of the major challenges faced by patients with RDs during the pandemic compared to those without RDs included the complete or partial interruption of medical therapy, delays in treatment, and greater enforced isolation.

The isolation policy Segmentation and Shielding in the UK [37], while essential for addressing pandemic challenges, was interpreted by some mastocytosis patients as an additional layer of segregation that created a ‘sense of otherness’ compared to non-RD patients [23]. Of LSD patients, 54% faced the lockdowns with a rediscovered appreciation for familial relationships, with only 30% of respondents in one study [7] having negative feelings about it, as opposed to 60% of the control group. The same positive approach to family members was found in other research, particularly for spouses [12] and in strengthening family units [6].

A great concern for RD patients was the interruption in public health services: 98.8% of Brazilian respondents experienced some interruption in health services [6], 60% of LSD patients refused to go to hospital in Italy to avoid the possibility of infection [7], and, during the first lockdown, all treatment for LSD patients was stopped in the UK [38]. A ‘negative healthcare effect’ was reported by 645 respondents in Hartinger’s research, and some linked it to difficulties in accessing RD-related care due to the lack of hospital space [12]. Similar results were found in Hong Kong, with 71% of respondents claiming to have experienced some delays in scheduled healthcare and 59% experiencing a complete halt [26].

Many RD patients experienced some form of negative reaction from people outside their groups, feeling disrespected or endangered in some way [12], including 87% of LSD patients [7], which was in line with the control group of this study.

As shown, for many RD patients, family became centrally important during the pandemic for managing new challenges and coping with associated difficulties. The documents highlighted a drastic opinion change of people with RDs towards ‘others,’ which further exacerbated the existing distinction between individuals with and without RDs, and institutions should take this into consideration to diminish the perceived differences between groups when other emergencies arise. A more unifying approach to communication could be used to level the differences in peoples’ perceptions while maintaining attention to vulnerable people in society.

#### 3.2.2. Families, Relatives, and Caregivers

FRCs experienced constant concern for their loved ones during the pandemic [8,9,14], such as parents trying to avoid the premature death of children [9] and coping with a lack of stable healthcare services. Caring for people with RDs during the pandemic was extremely taxing for people despite the high resilience they exhibited, and there were cases of parents arguing about who should care for the patient and who should work [9,24].

The absence of social support, such as daycare, which was previously taken for granted, placed caregivers in a new, more vexing and demanding situation [8]. Maintaining a good work–life balance became increasingly difficult [9], caregivers had to learn how to deal with new RD-related issues, and caregivers, like RD patients, experienced increases in loneliness, anxiety, and depressive moods [8].

RD patients often have healthy siblings, and, interestingly, Boettcher et al. [25] found that the healthy siblings of RD children often do not receive the same level of attention from caregivers. The pandemic exacerbated this situation, potentially leading to further isolation and challenges.

Based on these studies, it is possible to deduce that although FRCs were placed under extreme pressure by the pandemic, and they tried to cope resiliently, they constantly faced hardships. It should not be forgotten that, as for patients, their baseline quality of life differs [39] from that of people not concerned with RDs, and this fact should push the scientific community to protect them more.

#### 3.2.3. Organisations

The organisational issues were derived from only one source, but, nevertheless, we considered it important to provide a wider perspective on the concerns of various interested entities and the solutions they found during the pandemic.

During the pandemic, 80 RD-related organisations in the Asia–Pacific region responded to a survey designed to investigate their major concerns: 89% of the organisations were concerned about the impacts of COVID-19, and the focal aspects were ‘patients’ well-being’ in terms of physiological and physical support (33%); ‘organizational operations’ (23%) in the face of reduced funding, staff, and organisational activities; and ‘perceptions of and preparedness for COVID-19 and awareness of RDs’ (12%). Another aspect addressed by this study was how the organisations perceived COVID-19’s impact on RD patients. The answers focused on ‘healthcare access’ (43%) due to delays and cancellations of appointments, ‘social impacts’ due to isolation and a lack of social activities and learning (26%), ‘physical health impacts’ (14%) related to increases in infection vulnerability and insufficient protective equipment, ‘psychological impacts’ (11%) due to increases in anxiety and COVID-19-related burdens, and ‘financial impacts’ (6%) due to work layoffs and lockdown-related financial threats. Regarding the psychological impacts, it is notable, as the author stated, that differences in this aspect in other parts of the world could be due to cultural differences in the willingness to speak about mental health problems. To deal with this situation and help patients, Asia–Pacific organisations moved towards the digitalisation of previously face-to-face operations and considered the advantages of telehealth to be necessary for the future [18]. RD organisations as advocacy groups for people with RDs should be supported as far as possible by governments, because they are invaluable for generating connections and improving understanding between patients, their FRCs, and the healthcare system [40].

### 3.3. Interventions

Various forms of telehealth have been implemented around the world, as they were proved necessary for helping RD patients during the pandemic, and they are likely to remain necessary in the future [8,10,11,17,18,19,23,38]. Although telehealth is not suitable for everyone [19], it can certainly give RD patients, caregivers, and involved people without RDs more healthcare options. However, when introducing telemedicine, it is important to have a digital, language-accessible, official source of information that is accessible by and tailored to the specific problems and solutions for an individual RD or a group of similar RDs. Such a reputable source of information, which can be found in the professional psychologist, could alleviate anxiety and reduce confusion in already troubled people and guide them towards appropriate answers, solutions, or institutions. Psychological services should be activated for people with RDs, and professional education should be used to increase knowledge about RDs’ peculiarities. School programmes should be tailored as far as possible to account for the special needs of children with RDs, as has been done for Rett syndrome [41].

Professional psychologists were invaluable during the pandemic, and forms of long-distance therapy gave people with RDs help in developing better coping strategies and ways of using their resources. However, psychology researchers had different perspectives on this matter, which widened the understanding of the complexity of the pandemic implications for this population.

Regarding government measures, lockdowns and social distancing policies were used to decrease the health risks for vulnerable people, and, although often effective, they tended to result in psychological challenges [1,23,28].

### 3.4. Coping with the Pandemic

The data showed that 20% of Italian RD patients felt able to cope with the situation and use it as an opportunity for personal growth [7], and isolation reduced the hospitalisation of sick children [9], relieving parents of some of the added burden of the pandemic. Patients discovered inner resilience, began to study and cultivate hobbies [12], found time for family, discovered that families had more time for them, and/or found renewed meaning in relationships [7].

The fact that many people involved with RDs had comparable outcomes in COVID-19-related studies to people with no involvement with RDs suggests that despite the poorer baseline quality of life of the former, they had a potentially great ability to find solutions to difficult problems and to cope with drastic changes in lifestyle. This resilience, while being a truly important resource, should not be seen as a solution, but as a baseline for constructing a better psychosocial environment for these people so that their energy and ability can be spent on coping with other issues. The scientific community cannot afford to leave these people behind and allow them to become ‘once forgotten, always forgotten’ [42].

What was done to address the pandemic should not be forgotten, and the hard lessons the scientific community learned about the fragility and resilience of people involved with RD must be remembered. Much effort has gone into creating a better healthcare ecosystem in the broadest possible sense to connect professionals, people with RDs, and their families. The Irish National Health System [20] provided a good example to follow by creating a preliminary RD care intervention programme based on a multidisciplinary approach in collaboration with RD patient representatives and service users with 29 rare conditions.

## 4. Discussion

To summarise the analysed literature, patients and FRCs suffered a worsening of their conditions during the COVID-19 pandemic, as did people without RDs, but the major difference was the baseline quality of life of these two groups, with that of the first being clearly worse than that of the second. The studies critically highlighted the personal resources and resilience people with RDs employed to adapt to new situations and cope with the COVID-19 emergency. Telehealth became (and still is) an essential tool that needs to be improved and tailored to meet the special needs of some of the most fragile people in society and steadily implemented in healthcare plans. Families and caregivers of RD children showed severe increases in anxiety and psychological burdens, which should have been addressed with help from RD organisations, advocacy groups, and institutions. The lessons learned during the pandemic are vital for building better healthcare systems, both in preparation for future similar events and to provide high-quality service to people in need so that people with RDs are not forced to choose between compulsory therapies (often unavailable during pandemics) and safety from infection resulting from isolation.

Another major point is the lack of research in this field. People with RDs deserve more research because, like other sick people, they live with a constantly diminishing quality of life compared to healthy people. They have unique needs that must be addressed. Families and caregivers also deserve more attention from the scientific community, so that when calamities occur, the healthcare system and institutions are better prepared to help them.

To compare the results of this review with another suffering population, the same search conducted for this article was repeated by changing the keyword ‘rare diseases’ to ‘cancer.’ In this case, Google Scholar yielded 10,700 articles, Scopus yielded 546, and PubMed yielded 433. These numbers translate to a mean difference of roughly 5000% in published articles between those speaking about RDs and cancer (4931% in Google Scholar, 4963% in Scopus, and 6185% in PubMed), although the prevalence, it is important to remember, was the same. Confronting these data as a scientific community should prompt researchers to produce extensive research on the more than 420 million people with RDs (not counting their families, caregivers, and relatives) living their lives and coping with their vulnerability with (at least during the COVID-19 pandemic) fifty times less research conducted on their psychological well-being. To further highlight the importance of research on RDs and RD patients’ well-being, it is important to note that the prevalence of cancer in the population at any given age varies between 0.4% and 5.5% [43], making it a problem of the same magnitude as RDs in terms of people affected.

This review showed that lockdowns and social distancing resulted in people with RDs and their FRCs experiencing greater struggle. This result aligns with an extremely in-depth review of the impact of social distancing on vulnerable populations [21], but it further highlights how resourceful people with RDs can be in coping with challenging situations. These resources should always be considered by the psychological community when treating RD patients so as not to miss opportunities arising from their difficult experiences.

Similarly, the results of the psychological side of this review align with other studies on broader populations [1,21,44,45] showing an increase in mental struggles and psychological burdens, but the studies on RD patients have clearly shown the aforementioned resources of this population, again underling the importance of this aspect.

Other researchers have highlighted the importance of and the shift towards digitalised therapy both during and after the COVID-19 pandemic [46], and while we were not able to fully consider the implications of this shift, we can state that vulnerable people—in particular, those with restricted mobility—would benefit greatly from the implementation of such tools, which have already shown their utility for people with RDs.

Organisations concerned with RDs displayed similar resilience to RD patients both in the reviewed article and in another study conducted in Spain [47], which revealed their dynamism and proactivity in coping with decreased personnel and new challenges during the pandemic.

Most of the studies showed increased difficulty in accessing healthcare due to the system being saturated by COVID-19 patients, and although this was a problem for everyone, it was even more of a problem for people with RDs [48,49,50,51], showing the need for an improved healthcare system.

Considering the literature analysed overall, it must be pointed out that research on RDs is largely medical research, which has many implications. The first concerns the use of medical–diagnostic language when not considering genuine psychological diagnoses. It should be noted that the exclusive use of diagnostic labels to describe the psychological distress experienced by RD patients is inappropriate, and such distress should be labelled differently. The use of the same terms typically involved in clinical diagnoses (such as ‘anxiety’, ‘stress’, and ‘depression’) can lead to a further pathologizing of the experiences addressed by research, this time on a mental level, in turn further contributing to the medicalisation of their problems. More effective collaboration across disciplines, including psychology, sociology, and the human sciences, is vital for integrating their different points of view into the multidisciplinary approach necessary for the treatment of RDs. In addition, when there are no certain prospects of a cure, uncertainty afflicts people with RDs, and the psychological community should actively find ways to improve their psychological quality of life.

A multidisciplinary, organised, well-structured healthcare system is extremely necessary in the field of RD, as suggested by Ward [20]. Psychologists should invest time in trying to promote psychological well-being. Research could ease the burden on families, relatives, and caregivers of people with RDs and ultimately improve the quality of life of all those directly or indirectly involved with RD patients.

## 5. Conclusions

This systematic review has highlighted the efforts and struggles of people involved with RDs on many levels. Psychosocial burdens were experienced by most of the research participants (patients, FRCs, and organisations). Telehealth, although not an ultimate solution, has proven to be a fundamental tool in healthcare systems. Research centred on RDs should be a major concern for the scientific community, especially the psychosocial community, due to the lack of such research compared to research on other equally prevalent diseases. Research should be conducted with a broader approach and in dialogue with different disciplines to develop more efficient interventions in the future. Greater attention to this vulnerable segment of the population should be a government priority, possibly resulting in increased financing, a sensibilisation of public opinion on the struggles of people with RDs, and an increase in associated research. The varied results discussed in this paper can be considered a step towards realising the depth of struggles and strength shown by people affected with RDs and those who surround them. For psychologists, this article highlights the latent resilience of this population when facing crises, while showing the extremely delicate equilibrium that governs their lives. For clinicians engaged with RD patients, the research insights elucidated herein provide a holistic comprehension of the manifold challenges posed by the pandemic, highlighting not merely the medical adversities but also the deep-seated psychosocial repercussions. These findings suggest the need for more empathic and nuanced knowledge and the development of a communication strategy to enhance therapeutic alliances and outcomes. Leveraging the findings of this research would enable a profound understanding of the escalated anxiety and psychological encumbrances experienced by the families and caregivers of RD patients to be cultivated. This will necessitate an approach that transcends singular treatment modalities to encompass family-centric paradigms that acknowledge and address the increased anxiety and challenges confronting families and caregivers. Such an initiative could ostensibly steer clinicians to devise health strategies that are more inclusive, comprehensive, and holistic, actively integrating families and caregivers in the therapeutic trajectory. Moreover, clinicians can capitalise on the findings pertaining to the efficacy of telemedicine as a tool not only pivotal in the pandemic era but also possessing the propensity to redefine healthcare in the long term. The deployment of this technological advance could help clinicians construct strategies for providing more effective healthcare for RD patients, equipping them with the requisite apparatus to efficaciously steer their health pathways, particularly during crises. By fostering interdisciplinary collaborations and endorsing research that addresses the medical, psychological, and social needs of RD patients, we can forge a healthcare scheme that is genuinely responsive to the intricate needs of this demographic. This paper stresses the urgent need for reframing and championing a health strategy that is inclusive, empathetic, and attuned to the diverse and intricate needs of RD patients and their familial networks. Further exploration of the topics addressed in this paper, possibly with a focus on RD categories or a specific RD and taking a psychological perspective, should provide better future tools for professionals to treat RD patients more effectively, or at least to better comprehend the specificities involved.

### Limitations

The lack of English literature on the psychological aspects of the pandemic for people with RDs was the first limitation of this systematic review but also its strongest motivation. Specifically, we identified a lack of literature focusing on clinical psychology and its potential contribution to counselling people with RDs. Additionally, the few studies available have mainly employed medical approaches and have been very scattered because they have been spread across many different RDs. All of these factors made this work difficult, but they testify to the urgent need to gather information and develop knowledge that can be useful to doctors, psychologists, and other health professionals, while, at the same time, representing a call to action to gather more data, more information, and more experiences from people affected by RDs to develop a better adaptation to life in the future.

## Figures and Tables

**Figure 1 ijerph-20-06863-f001:**
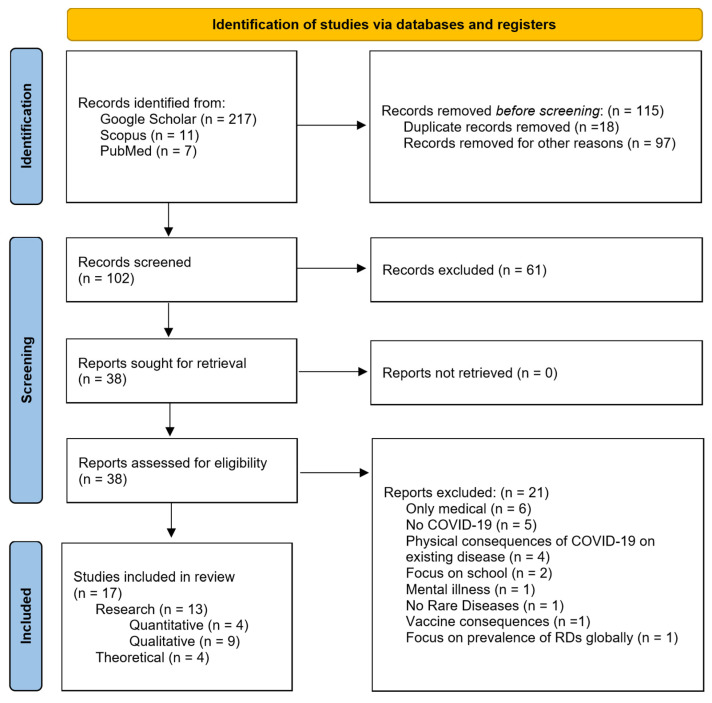
Flowchart of the literature search.

**Table 1 ijerph-20-06863-t001:** Document synthesis.

Author, Year, Country	Research Focus, Number of Participants, Control Group (Y/N) Findings	Document Nature, Researcher Background
Marano, G.; Gaetani, E.; et.al, 2020 [5]; Italy	Interventions for hereditary haemorrhagic telangiectasia patients during COVID-19 Telehealth and multidisciplinary approach needed for effective interventions	Paper, review, psychology, medicine
Schwartz, I.V.D.; Randon, D.N.; et.al, 2021 [6], Brazil	COVID-19 impact on RD patients, n = 1466, Y Isolation helped contain mortality; telemedicine was an essential tool to face healthcare system flaws	Paper, research, medicine
Fiumara, A.; Lanzafame, G.; et.al, 2020 [7], Italy	COVID-19 impact on lysosomal storage disorder patients, n = 15, Y LSD patients’ needs to adhere to therapy schedule; the pandemic strengthened familiar relationships and worsened social ones	Paper, research, medicine
Rihm, L.; Dreier, M.; et.al, 2022 [8], Germany	COVID-19 impact on families of RD children, n = 149, N Caregivers of RD children endured great distress during the pandemic; family-centred psychological support is needed to help caregivers coping with emergency situations	Paper, research, psychology
Stewart, A.; Smith, C.H.; et.al, 2021 [9], England	COVID-19 impact on parents of oesophageal atresia/tracheo-oesophageal fistula patients, n = 65 + 6, N Telehealth, as necessary as it is, needs improvement to address specific needs; caregivers’ experiences were varied and, as such, so should be the institutions’ responses	Paper, research, medicine, language therapy
Smyth, D.; Hytiris, M.; et.al, 2023 [10], England	COVID-19 impact on osteogenesis imperfecta patients, n1 = 110, n2 = 124, N Remote consultation played an important role in managing the emergency and should be implemented in the healthcare system; there were differences in results between T1 and T2 of the research, highlighting differences in patients’ perceptions and needs during the pandemic	Paper, research, medicine
Marano, G.; Gaetani, E.; et.al, 2021 [11], Italy	Interventions for interstitial cystitis/bladder pain syndrome patients during COVID-19 Web-mediated counselling could help patients and their caregivers to alleviate distress with a tailor-made approach; a multidisciplinary approach is advised	Paper, review, psychology, medicine
Hartinger, A.V., 2022 [12], USA	COVID-19 impact on RD patients, n = 759, N COVID-19 pandemic had a great impact on RD patients, who were demonstrated to be resilient, and it strengthened familiar relationships; hope for needed changes grew thanks to the pandemic highlighting healthcare system flaws	Thesis dissertation, research, psychology
Fjermestad, K.W.; Orm, S.; et.al, 2022 [13], Norway	COVID-19 impact on RD patients, n = 58, Y RD patients showed worse indexes in SLC-5 and CAS scale compared with non-RD people; anxiety management is advised for professionals working with RD patients	Paper, short report, psychology
McMullan, J.; Crowe, A.; et.al, 2020 [14], England	COVID-19 impact on caregivers, n = 165, N Caregivers experienced high psychological burden during the COVID-19 pandemic; a tailored approach is advised to better help alleviate the burden on this population	Conference, research, psychology
Chung, C.C.Y.; Ng, Y.N.C.; et.al, 2021 [17], Asia–Pacific region	COVID-19 impact on RD-related organisations RD-related organisations experienced decrease in manpower and funding; digitalization was needed to cope with the situation and manage to help those in need	Paper, research, medicine
Scherr, J.F.; Albright, C.; et.al, 2021 [18], U.S.A.	Interventions for Batten disease during COVID-19 Telehealth is an important tool to address special needs of RD patients; a tailor-made approach should be implemented for better results	Paper, review, medicine
Chowdhury, S.F.; Sium, S.M.A.; et.al, 2021 [19], Global	Interventions for patients and research during COVID-19 RD people endured a worsening of an already difficult situation; institutions should focus on RD people’s special needs and provide adequate care and access to therapy	Paper, review, medicine
Chaplin, C., 2021 [23], England	COVID-19 impact on mastocytosis patients, n = 3, N Telemedicine was one of the only aids for patients enduring new challenges and isolation during the pandemic; resilience emerged as an important aspect to adjust to a new normality; a lack in non-RD people’s understanding of the challenges is highlighted	Paper, research, medicine
Martin, J.; Robertson, K.; et.al, 2022 [24], England	COVID-19 impact on parents of rare neurogenetic conditions children, n = 11, N COVID-19 pandemic highlighted some flaws in the healthcare system, giving insight into how to improve the services; a multidisciplinary approach is suggested to offer appropriate support; children with neurogenetic disorders were negatively impacted by the pandemic	Paper (peer review pending), research, psychology
Boettcher, J.; Nazarian, R.; et.al, 2021 [25], Germany	COVID-19 impact on siblings of children with rare congenital surgical diseases, n = 81, Y Not only patients but also siblings of RD children have been largely impacted by the COVID-19 pandemic; healthy siblings could have experienced a lack of attention to their needs; greater attention from professionals and families alike is advised	Paper, research, medicine
Chung, C.CY.; Wong, W.HS.; et.al, 2020 [26], China	COVID-19 impact on RD patients RD patients suffered from both higher risk of COVID-19 infection and difficulties in accessing healthcare services, increasing psychological burden	Paper, research, medicine

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
