# Peer review of "Caring for People with Rare Diseases: A Systematic Review of the Challenges of, and Strategies for Dealing with, COVID-19"

_ijerph, 2023, doi:10.3390/ijerph20196863_

Round 1
Reviewer 1 Report
The review by Elena and colleagues investigated the impact of the COVID-19 pandemic on rare diseases. The manuscript is interesting; however, I have provided few gentle comments and suggestions below.
Comments/suggestions and typos
Title:
COVID-19 not COVID 19.
Abstract:
COVID-19, not Covid-19; the word ‘COVID-19’ need to be capitalized in the manuscript.
The abstract can be improved; the abstract should summarize the research background, the key methods, results and conclusions.
Line 24: You should not include references in the abstract.
Introduction:
Line 59: ‘only on Scholar’ do you mean ‘only on Google Scholar’. This phrase is not clear. The same is in line 95.
Results
Line 142: ‘the psychological e psychosocial’ ‘the psychological’.
Line 151: “Tab.1’ ‘Table 1),
Table
The table legend (title) should be placed above the table (not within) in left alignment, and please write the word ‘Tab.1’able 1’.
Author Response
Response to Reviewer 1 Comments
Thank you very much for taking the time to review this manuscript. Please find the detailed responses below and the corresponding revisions/corrections highlighted/in track changes in the re-submitted files .The review by Elena and colleagues investigated the impact of the COVID-19 pandemic on rare diseases. The manuscript is interesting; however, I have provided few gentle comments and suggestions below.
Comments/suggestions and typos
Title:
COVID-19 not COVID 19.
Modified
Abstract:
COVID-19, not Covid-19; the word ‘COVID-19’ need to be capitalized in the manuscript.
Modified
The abstract can be improved; the abstract should summarize the research background, the key methods, results and conclusions.
The abstarct was modified following the suggested directions:
The COVID-19 pandemic took a toll on the lives of everyone, and patients with rare diseases had to pay an even higher price. This systematic review explores the impact of the COVID-19 pandemic on individuals with rare diseases from a psychological perspective. Using the Preferred Reporting Items for Systematic Reviews and Meta-Analyses (PRISMA) guidelines, articles were sourced from Google Scholar, Scopus, and PubMed databases, focusing on "COVID-19", "Psychology", and "Rare diseases". Seventeen primary articles (mainly from Continental Europe) and ten of secondary interest were identified. Results show a notable focus on the challenges faced by rare disease patients, including increased anxiety, stress, and depression. There's a substantial difference in the volume of research on rare diseases compared to prevalent diseases like cancer, indicating a significant gap in knowledge. The review underscores the heightened vulnerabilities of rare disease patients during the pandemic, especially regarding the quality of life, and emphasizes the importance of telehealth as a critical intervention. Conclusions highlight the pressing need for multidisciplinary research and more robust healthcare systems to cater to rare disease patients' unique challenges, representing between 3.5 and 5.9% of the global population.
Line 24: You should not include references in the abstract.
References have been deleted
Introduction:
Line 59: ‘only on Scholar’ do you mean ‘only on Google Scholar’. This phrase is not clear. The same is in line 95.
Modified
Results
Line 142: ‘the psychological e psychosocial’ ‘the psychological’.
Modified
Line 151: “Tab.1’ ‘Table 1),
Modified
Table
The table legend (title) should be placed above the table (not within) in left alignment, and please write the word ‘Tab.1’able 1’.
Modified
Reviewer 2 Report
Thank you for the opportunity to review your article.
I want to start by saying that I'm an English-speaker who resides in North America. I had a challenging time reading through your article. There is a need to go back and correct several grammatical mistakes. Here are just a few that highlight the ones throughout your paper:
Line 33: Add a comma between "recognition" and "many governments" -- "With this recognition, many governments..."
Lines 84 to 85: This is an incomplete sentence with which you started your Materials and Methods section. Needs to be re-written to something like "The goal of this study was to study the COVID-19 pandemic outcomes in the life and psychological well-being of people with rare diseases and all the social actors revolving around them."
Lines 97-99: Break these sentences up to something like: "These searches were performed on April 12, 2023, with an updated search on May 23, 2023. In the repeat search in May, the Google Scholar output counted 220 articles, while the other two search engines remained unchanged."
Throughout your article, please keep the dates consistent with how you write them. In the United States, we write them like "May 23, 2023." If you are going to write them like "the 11th of February, 2020", then do that consistently throughout your article.
On line 39, you write "people suffering from RDs" but hadn't yet indicated that RD stands for rare diseases in your main article.
I found it confusing in your introduction that you seem to first reference rare diseases in general but then bring up cancer in particular. I would really like to see your introduction rewritten so that it flows better and more in line with what you are actually researching. Also define terms like "social actors," etc in your introduction, as these are not universally known.
In line 57, you state "The search yielded 217 articles..." What search? You haven't yet indicated that a search was being done. Again, rewrite the intro to flow better.
In line 79, I would choose a different way to describe these individuals rather than "special need people." Maybe, individuals diagnosed with a rare disease?
The conference mentioned as one of your search results, please describe. Was this a poster presented at a conference or a talk?
I had a hard time following your results, mostly due to broken English and not a consistent flow of your results. Please rewrite these to flow better and be grammatically correct.
I don't think you need to add sentences like what you did to begin your Discussion section. If you make this article flow better, the reader is going to know that without you saying it.
Finally, for your conclusion, yes, it would be nice to make this a Government priority, but what would that look like? Also, as a clinician reading this, how do these search results help me in working with patients?
I addressed this above in the general comments section, so please review. There is extensive editing needed to have your article be grammatically correct.
Author Response
Response to Reviewer 2 Comments
Thank you for the opportunity to review your article.
I want to start by saying that I'm an English-speaker who resides in North America. I had a challenging time reading through your article. There is a need to go back and correct several grammatical mistakes. Here are just a few that highlight the ones throughout your paper:
Thanks for the pointers, the article has been linguistically reviewed by native speaker academic reviewers. After revision, the lines indicated no longer correspond.
Line 33: Add a comma between "recognition" and "many governments" -- "With this recognition, many governments..."
Comma has been added
Lines 84 to 85: This is an incomplete sentence with which you started your Materials and Methods section. Needs to be re-written to something like "The goal of this study was to study the COVID-19 pandemic outcomes in the life and psychological well-being of people with rare diseases and all the social actors revolving around them."
The sentence has been completed as suggested
Lines 97-99: Break these sentences up to something like: "These searches were performed on April 12, 2023, with an updated search on May 23, 2023. In the repeat search in May, the Google Scholar output counted 220 articles, while the other two search engines remained unchanged."
The sentence has been split as suggested
Throughout your article, please keep the dates consistent with how you write them. In the United States, we write them like "May 23, 2023." If you are going to write them like "the 11th of February, 2020", then do that consistently throughout your article.
Thank you, we have adopted the following mode throughout the article:
These searches were performed on April the 12th 2023 and the search updated the May the 23rd 2023.
On line 39, you write "people suffering from RDs" but hadn't yet indicated that RD stands for rare diseases in your main article.
On line 40 (page 1-2) you can find now “from rare diseases (henceforth RD)”
I found it confusing in your introduction that you seem to first reference rare diseases in general but then bring up cancer in particular. I would really like to see your introduction rewritten so that it flows better and more in line with what you are actually researching. Also define terms like "social actors," etc in your introduction, as these are not universally known.
The reference to cancer was removed in the introduction as it was confusing.
In the body of the article, data are given that show how rare diseases as a whole have incidence figures comparable to cancer, and yet are much less studied.
In line 57, you state "The search yielded 217 articles..." What search? You haven't yet indicated that a search was being done. Again, rewrite the intro to flow better.
In the introduction, all the steps involved in the research were better specified
In line 79, I would choose a different way to describe these individuals rather than "special need people." Maybe, individuals diagnosed with a rare disease?
The expression was modified
The conference mentioned as one of your search results, please describe. Was this a poster presented at a conference or a talk?
It is the abstract of a presentation given at a conference, it refers to research in psychology on the topic. a reference is available in table 1, cit number 14
I had a hard time following your results, mostly due to broken English and not a consistent flow of your results. Please rewrite these to flow better and be grammatically correct.
I don't think you need to add sentences like what you did to begin your Discussion section. If you make this article flow better, the reader is going to know that without you saying it.
Thank you, the results have been rewritten and linguistically revised. The sentence at the beginning of the discussion has been removed.
Finally, for your conclusion, yes, it would be nice to make this a Government priority, but what would that look like? Also, as a clinician reading this, how do these search results help me in working with patients?
In the current version of the article, various ideas have been added on how to make this issue a priority for the government and to guide health policy. various considerations for the advancement of clinical practice arising from the review findings have also been added.
Reviewer 3 Report
The World Health Organization has already warned about the psychological impact of COVID-19 and the impact of symptoms on mental health problems and mental disorders. For people with rare diseases, the pandemic has brought great mental and emotional challenges that need to be addressed. The need for psychological support is essential to cope with these situations and I think it is a very relevant topic.
The title states the research problem in clear and understandable words, but is too long (reduce to a maximum of 15 words).
The abstract gives the basic content, describes the aim, methodology and results, the conclusions should include the importance of psychology in the intervention. Abbreviations and references should be removed from the abstract.
The introduction identifies some aspects of what is known about the issue, although it does not go into much depth. I miss the rationale for the importance of psychological support. The references in the introduction are those chosen for the review, they should be references that explain the need for the review. The aim is ambitious and not fully addressed.
The structure of the manuscript is correct, although the introduction, methods and discussion sections are underdeveloped.
The manuals currently used in psychology are DSM-V and ICD-11. And they should be included in the references.
The analysis of the results on organisations includes only one article, which discusses the relationship with other parts of the world in a hypothetical way. However, there are several studies on organisations in other countries that could help to concretise this result, and two examples from Peru and Spain are added: Vara Graña, Carlos Giusseppe. Impact of the COVID-19 pandemic on people living with rare diseases in Peru in 2021. (2022). Y Guerra, José Manuel, Gema Esteban-Bueno and Juan R. Coca. The impact on the activity of rare disease organisations and families during the COVID-19 closure in Spain. Acciones e Investigaciones Sociales 42 (2021).
Acronyms should be explained the first time they are used: WHO, RDs....
COVID-19 is always written in capital letters; in the text it is sometimes written in capital letters and sometimes in lower case. Table 1, use the full name in the text and in the table heading.
The search strategies are well described, but the inclusion and exclusion criteria are not clear, so I do not understand why theoretical or review papers are included.
I miss the use of a quality system to filter the articles, for example the CASPe CRITICAL READING PROGRAMME, or the specific criteria of PEDro, or any other system that helps to establish the transparency of the results and findings of the evidence synthesis. And the references selected in the review should be marked in some way in the final list.
In the section on interventions, the role of the psychological professional is not specified, nor are prevention and recovery efforts substantiated.
In the discussion, the results are discussed in relation to several important points, but then not compared with the results of other similar published studies.
Limitations and how they affect the conclusions in relation to the objective are not discussed. Data on rare diseases compared with cancer are discussed twice (introduction and discussion).
The conclusions are consistent with the content of the objectives.
The number of references is adequate, although it would be good if the articles selected in the review were identified in the list.
Author Response
Responses to Reviewer 3
The World Health Organization has already warned about the psychological impact of COVID-19 and the impact of symptoms on mental health problems and mental disorders. For people with rare diseases, the pandemic has brought great mental and emotional challenges that need to be addressed. The need for psychological support is essential to cope with these situations and I think it is a very relevant topic.
The title states the research problem in clear and understandable words, but is too long (reduce to a maximum of 15 words).
Thank you, the title was shortened.
Caring for people with rare diseases: challenges and strategies facing the COVID-19, a systematic review
The abstract gives the basic content, describes the aim, methodology and results, the conclusions should include the importance of psychology in the intervention. Abbreviations and references should be removed from the abstract.
All suggestions have been integrated into the current version of the abstract
The introduction identifies some aspects of what is known about the issue, although it does not go into much depth. I miss the rationale for the importance of psychological support. The references in the introduction are those chosen for the review, they should be references that explain the need for the review. The aim is ambitious and not fully addressed.
A fuller discussion was made possible by citing recently published articles that had not been included in the review because they did not meet the selection requirements. For example, articles 47 and 48, although not in English and, as such, not eligible for review, were cited to discuss organisations or suffering people, just as some articles (49 to 51) were cited to draw parallels with other types of illnesses (AIDS, rheumatic diseases, cancer); only a small part of the available literature was cited here as an example for a different perspective, so as not to digress too much. Articles 1, 44 and 45 were selected to discuss the psychological burden from different perspectives, while Article 46 deals with the digitisation process. Most of the articles chosen in this section were published in 2023 and are reviewed; this choice was made in order to compare literature of the same type (written after the pandemic) and possibly addressing the problem from a broad perspective. We decided to compare our work with others not necessarily focused on rare diseases to further extend and strengthen our findings and to address the lack of English literature on the psychological aspect of rare diseases during the pandemic.
The structure of the manuscript is correct, although the introduction, methods and discussion sections are underdeveloped.
All sections have been deepened
The manuals currently used in psychology are DSM-V and ICD-11. And they should be included in the references.
The diagnostic manuals were cited and included in the references
The analysis of the results on organisations includes only one article, which discusses the relationship with other parts of the world in a hypothetical way. However, there are several studies on organisations in other countries that could help to concretise this result, and two examples from Peru and Spain are added: Vara Graña, Carlos Giusseppe. Impact of the COVID-19 pandemic on people living with rare diseases in Peru in 2021. (2022). Y Guerra, José Manuel, Gema Esteban-Bueno and Juan R. Coca. The impact on the activity of rare disease organisations and families during the COVID-19 closure in Spain. Acciones e Investigaciones Sociales 42 (2021).
Thank you for suggesting these articles. They are very relevant and thorough. They have been added to the discussions. It was not possible to include them in the review items as they are unfortunately not available in English.
Acronyms should be explained the first time they are used: WHO, RDs....
All acronyms were made explicit when first used
COVID-19 is always written in capital letters; in the text it is sometimes written in capital letters and sometimes in lower case.
It was decided to always write COVID in capital letters
Table 1, use the full name in the text and in the table heading.
Done
The search strategies are well described, but the inclusion and exclusion criteria are not clear, so I do not understand why theoretical or review papers are included.
The current version of the paper better specifies the inclusion end exclusion criteria:
The screening process consisted in first and second author independently going through abstract and text of the first 102 (47%) articles. After the first screening a discussion between the two authors was centered around the inclusion criteria and considering the focus of this review, it was decided to exclude articles that were focused only on the medical aspects of the relation between COVID-19 and RDs. It was decided to focus on articles only written in English and decided to keep documents from all around the world, not to limit to any particular RD and to consider equally salient the medical’s point of view, the patient’s, the relatives/caregivers’ and the RDs associations. In this phase 61 records were excluded due to inconsistency with the objectives of the paper.
I miss the use of a quality system to filter the articles, for example the CASPe CRITICAL READING PROGRAMME, or the specific criteria of PEDro, or any other system that helps to establish the transparency of the results and findings of the evidence synthesis.And the references selected in the review should be marked in some way in the final list.
The criteria for the selection of qualitative articles were also specified according to the methodology of the Prism Statement (see both text and Figure 1. Flow Diagram)
In the section on interventions, the role of the psychological professional is not specified, nor are prevention and recovery efforts substantiated.
This theme was added, see text added from line 399 onwards.
In the discussion, the results are discussed in relation to several important points, but then not compared with the results of other similar published studies.
New quotations (47,48,20) now appear in the discussions
Limitations and how they affect the conclusions in relation to the objective are not discussed.
A section called 'limitations' was inserted after the conclusions
Data on rare diseases compared with cancer are discussed twice (introduction and discussion).
Cancer data were removed in the introduction
The conclusions are consistent with the content of the objectives.
The number of references is adequate, although it would be good if the articles selected in the review were identified in the list.
Revised articles are available in the table. they appear in association with the article's reference number.
Thank you for all the suggestions offered, which we found very useful.
The authors